# Selenium as a Bioactive Micronutrient in the Human Diet and Its Cancer Chemopreventive Activity

**DOI:** 10.3390/nu13051649

**Published:** 2021-05-13

**Authors:** Dominika Radomska, Robert Czarnomysy, Dominik Radomski, Anna Bielawska, Krzysztof Bielawski

**Affiliations:** 1Department of Synthesis and Technology of Drugs, Medical University of Bialystok, Kilinskiego 1, 15-089 Bialystok, Poland; dominika.radomska@umb.edu.pl (D.R.); d.radomski@post.pl (D.R.); kbiel@umb.edu.pl (K.B.); 2Department of Biotechnology, Medical University of Bialystok, Kilinskiego 1, 15-089 Bialystok, Poland; aniabiel@umb.edu.pl

**Keywords:** selenium, cancer, prevention, cancer chemoprevention, molecular mechanisms, micronutrients, essential nutrients, bioactive nutrients, nutrient sources, COVID-19

## Abstract

This review answers the question of why selenium is such an important trace element in the human diet. Daily dietary intake of selenium and its content in various food products is discussed in this paper, as well as the effects of its deficiency and excess in the body. Moreover, the biological activity of selenium, which it performs mainly through selenoproteins, is discussed. These specific proteins are responsible for thyroid hormone management, fertility, the aging process, and immunity, but their key role is to maintain a redox balance in cells. Furthermore, taking into account world news and the current SARS-CoV-2 virus pandemic, the impact of selenium on the course of COVID-19 is also discussed. Another worldwide problem is the number of new cancer cases and cancer-related mortality. Thus, the last part of the article discusses the impact of selenium on cancer risk based on clinical trials (including NPC and SELECT), systematic reviews, and meta-analyses. Additionally, this review discusses the possible mechanisms of selenium action that prevent cancer development.

## 1. Introduction

In the 19th century, Swedish chemist Jöns Jakob Berzelius discovered selenium while researching a method of obtaining sulfuric acid from pyrites containing iron sulfide [1]. Initially, it was regarded as a toxic element, and its role as an essential trace element in the body was not established until 150 years after its discovery, in 1957 [2]. This was determined by Klaus Schwartz and Calvin Foltz, who considered it a substance responsible for preventing lesions in blood vessels, the liver, and/or muscles in rats and chickens. Since then, its impact on the human body and the mechanisms involved have been thoroughly studied. An important aspect was linking selenium deficiency, as well as its excess, with disease occurrence and its impact on cancer development risk [3].

The aim of this review is to discuss the biological activity of selenium, i.e., its impact on the proper functioning of the body through various selenoproteins, which perform many important physiological roles [4]. They are responsible, among other things, for immunity, fertility, the aging process, and the management of thyroid hormones, but first and foremost their main function is maintaining a redox balance in cells, which also affects the aforementioned functions [5,6,7,8,9]. Moreover, the article addresses the issue of appropriate selenium intake in the diet and the effects of its deficiency and excess in the body, and, very importantly, its association with cancer risk. The number of new cancer cases and cancer mortality are a worldwide problem [10]. For this reason, possible prophylactic measures are being sought to prevent cancer development and thus reduce the number of new cases and deaths. The many studies associating cancer risk with selenium ingestion (and its levels in the body) indicate that this micronutrient may reduce this risk, which would be indicative of its chemopreventive properties [4]. The possible mechanisms responsible for this activity are also discussed in the last part of this review.

## 2. Dietary Intake of Selenium

### 2.1. Selenium in the Diet

Selenium (Se) occurs as a trace element in the human body, performing a number of important biological functions, and is essential for life and health [11]. The main natural source of Se is food, but Se content varies. This is determined by the geographical location, soil quality in terms of Se concentration, and how much it is accumulated in plants. The climate and the way the food is cultivated and bred, and how we prepare it for consumption, are also important factors [3,12]. The richest sources of Se in the diet are Se-yeast (Se-enriched yeast), nuts, cereals, organ meats, fish, and seafood [13]. Brazil nuts (Bertholletia excelsa, family Lecythidaceae), a plant from the Amazon, have the highest known Se content among non-Se-enriched food [14]. Table 1 gives the Se content in some examples of food products in our diet.

This microelement can exist in food in both chemical forms—as inorganic and organic compounds [24]. Mostly, Se is found in the form of organic compounds as selenomethionine (SeMet) and selenocysteine (Sec) in plant [25] and animal [26,27] tissues. However, the form of Se intake can affect its bioavailability in the body. Studies show that organic Se (e.g., SeMet) is more bioavailable than its inorganic compounds such as selenates or selenites [27,28,29]. Interestingly, the bioavailability of this element in the body is also modified by race. Richie et al. observed that in whites it is significantly higher than in blacks [30]. Nevertheless, the most important thing is a healthy and balanced diet that contains all the nutrients at the right levels. As far as Se and its recommended daily dose are concerned, it depends on the local content of this element in the cropland, as this affects its content in the food [31,32,33]. According to World Health Organization (WHO) standards, the recommended daily dose of Se for adults is 55 µg/day [13,34,35], while the maximum tolerable adult intake without side effects is set at 400 µg/day [13]. Of course, the requirement for Se depends on age and gender [13], as shown in Table 2.

It is very important to maintain an adequate level of Se—both deficiency and excess can be dangerous for human health. What is important is that this microelement has a narrow range of safety [36,37,38], so one should ensure its optimal level in the body. For biomarkers of selenium status in the body, measuring the activity of selenoproteins such as glutathione peroxidase 3 (Gpx3) and selenoprotein P (SelP, Sepp1) in plasma proves to be valuable because a decrease of their activity indicates directly a deficiency of this trace element [39,40,41]. The optimal values of Se in plasma have been estimated at 90–120 µg/L (Figure 1), which is sufficient to saturate selenoproteins in this liquid blood component [40]. The limit below which a deficiency of this element is found is set at 85 µg/L [5,42], while in Poland, its concentration in blood plasma is below this value and is about 50–55 µg/L [43], which would suggest deficiencies in the inhabitants of this country. It is worth emphasizing the fact that Cui et al. [44] observed that low serum Se concentration is associated with a higher risk of prostate cancer. Additionally, there are reports that Se also plays a key role in the prevention of other cancers [45], i.e., lung [46], breast [47], bladder [48], gastric [49,50,51], thyroid [41,52,53], and esophageal [50,51], and at a dose of 100 to 200 µg/day reduces genetic damage [1]. Furthermore, in 1994, a clinical trial conducted by Kiremidjian-Schumacher et al. [54] concluded that sodium selenite (Na_2_SeO_3_) supplementation at a dose of 200 µg/day for eight weeks significantly increased the activity of T lymphocytes and NK cells.

### 2.2. Absorption of Se and Its Transport in the Body

As mentioned earlier, Se is taken with the diet in the form of organic (SeMet and Sec) and inorganic compounds (selenites, selenides) [55]. The absorption of these compounds mostly occurs in the duodenum and cecum, and their absorption mechanism depends on the chemical form of ingested Se. Inorganic compounds are absorbed by simple diffusion (selenites) or by secondary active transport, the so-called cotransport (selenides). In turn, organic compounds (SeMet, Sec, methylselenocysteine (MSC)) are absorbed by active transport in the same way as amino acids. For example, SeMet is absorbed via active transport using a sodium-dependent pump similarly to amino acids such as methionine [56,57].

After absorption of selenite and other Se compounds into the bloodstream, their rapid and selective uptake by red blood cells occurs. Subsequently, organic Se compounds undergo a reaction involving γ- or β-lyase, and selenite is reduced by glutathione (GSH) and glutathione reductase (GR). These reactions generate selenide, which is metabolized by selenophosphate synthetase 2 to selenophosphate—in this form, Se is transported to the liver [57,58]. After Se uptake by the liver, it is included in the synthesis of Sepp1 [55,58]. Sepp1 is the selenoprotein responsible for the transport of Se to peripheral tissues and organs because it has many Sec residues [57,58]. Additionally, there are also reports that Se can bind to low and very low density lipoproteins (LDL and VLDL) and α- and β-globulins and can also be transported in this way [57].

### 2.3. Selenium Deficiency

The problem of Se deficiency affects about 0.5–1 billion people in the world due to its insufficient consumption. This factor depends on the geographical area and correlates with the low content of this microelement in the soil. Regions with low soil Se content, include parts of the Congo, large parts of China, central Serbia, and, before 1984, Finland, which currently uses Se-enriched fertilizers [59]. The best known endemic diseases caused by Se deficiency, otherwise known as “geochemical diseases”, are Keshan and Kashin-Beck. The first cases of Keshan disease (KD) were discovered over 80 years ago in Keshan County (Heilongjiang Province, north-eastern China) [60]. The disease was characterized by congestive cardiomyopathy and affected children aged 2–7 years and women of reproductive age [61], whereas Kashin-Beck disease (KBD) occurs most frequently in Tibet [62] and begins in childhood around age five [63,64]. KBD is characterized by osteoarthritis, leading to the degradation of cartilage in the upper and lower limbs [63]. A too low level of this microelement in the body is also responsible for infertility in men, impaired fetal development [65], and increased risk of suffering from asthma (in the case of asthma, this is associated with a reduction in antioxidant defense, among other things, and a decrease in Gpx activity) [66]. There is also evidence that Se deficiency weakens the immune system [2] or the proper functioning of the nervous system [67]. Conner et al. [68], in their clinical study, observed that too-low Se levels in serum were associated with worse mood and increased depressive symptoms in young adults. For many years, scientists have been searching for answers as to whether Se affects the HIV viral load and AIDS progression. In 2019, a systematic review of RCTs was published, which concluded that Se has no effect on suppressing or reducing the viral load of HIV, but that there is clinical evidence that it is possible to prolong AIDS progression by supplementation with this element. Despite this, no definitive conclusions can be drawn due to their heterogeneity [69].

### 2.4. Selenium Overdose

Se poisoning occurs very rarely and is the result of excessive supplementation or a diet rich in products with a high content of this microelement. The consumption of “Coco de Mono” (*Lecythis ollaria*) nuts, which accumulate huge amounts of Se (7–12 g Se/kg of dry matter), caused acute Se poisoning in Venezuela [70]. The excess Se intake was associated mainly with the geographical area, whose soils are characterized by its high content [71]. An example of such dependence is Enshi (Hubei Province, South China), where there is a high content of Se in the soil—the inhabitants of this area had symptoms of the toxic effects of Se on the body because they had a dietary intake of over 850 µg Se/day [13]. On the other hand, it is also worth noting that taking lower doses of Se in the order of 300 µg/day may have adverse effects on the body. In the Danish Prevention of Cancer by Intervention with Selenium (PRECISE) trial, it was observed that long-term intake of 300 µg Se/day in the form of Se-enriched yeast increased mortality in the study population. For this reason, such a high daily supplementation or intake of Se should be avoided [72].

The toxicity of this element depends on many factors—its chemical form, ingested dose, interactions with other dietary components, and the physiological condition of the body [73]. As far as Se compounds are concerned, its inorganic forms exhibit higher toxicity than when they are in the form of organic compounds. This is because inorganic Se (selenides, selenites) has a prooxidant effect on thiols (GSH), producing free oxygen radicals, while methylated forms (organic) are less toxic due to their easier excretion [19]. Symptoms of acute Se poisoning are rather non-specific and therefore may cause problems in diagnosis. These include hypotension, rapid heartbeat (tachycardia), neurological disorders, fever, dry cough, and pulmonary edema. In both cases of toxicity—acute and chronic—anemia, gastrointestinal disorders, salivation, and blindness occur. In contrast, chronic Se toxicity, otherwise known as selenosis, is characterized by nail fragility, hair loss, skin lesions, joint pain, tooth decay, and a specific garlic odor in the exhaled breath (presence of volatile compound—dimethyl selenide) [13,74,75]. Excessive doses of Se also cause endocrine disorders in the synthesis of thyroid hormones, growth hormones (GH), and insulin-like growth factors (IGF-1) [76]. It is also worth emphasizing the link between Se and type 2 diabetes mellitus (T2DM). A meta-analysis of the observational studies conducted by Kim et al. in 2019 [77] showed that increased Se intake increases the risk of T2DM. Very importantly, this meta-analysis found that the odds of developing T2DM in people with high Se levels are approximately twofold higher than the odds in people with lower or optimal levels of this trace element. Se is not only found in the soil, water, and food; poisoning with its compounds may also occur as a result of inhalation (e.g., highly toxic H_2_Se), so its maximum concentration in the air should be less than 0.2 mg/m^3^. Symptoms of acute inhalation intoxication are chemical pneumonia, lung hemorrhage and edema, and bronchiolitis, as well as extrapulmonary effects—nausea, headaches, and eye irritation [15,78].

## 3. Biological Activity of Selenium

### 3.1. Selenoproteins in the Human Body

Se is a micronutrient that plays many key roles in maintaining homeostasis in organisms. It is worth emphasizing that Se as an element does not exhibit biological effects on the human body—only its presence in enzymes and proteins (e.g., selenoproteins) determines its activity [79]. It is present in the form of selenocysteine in the active centers of Se-dependent enzyme molecules, while proteins having at least one selenocysteine residue in their structure are called selenoproteins. Selenocysteine (Sec) is classified as the 21st protein amino acid, and its structure is similar to that of cysteine, with which it differs only in the presence of an Se atom instead of a sulphur atom in its structure. The residues of this amino acid are inserted into the proteins in the cotranslation process because it is coded by the UGA codon, which usually means the codon “STOP” and terminates protein synthesis [80,81,82,83]. In the human genome, 25 genes that encode selenoproteins have been discovered [81,84]. The products of expression of these genes are found in many tissues of the body and perform various functions. Three main families of Se-containing enzymes have been classified: glutathione peroxidases, thioredoxin reductases, and iodothyronine deiodinases, and other single selenoproteins [45,55,81,83], as shown in Figure 2, which also indicates their localization in the human body. Their expression is dependent on Se concentration. Too low or too high levels of this element regulate the transcription of these macromolecules but do not affect the rapidity of this process [2,83]. Differences in the expression of the genes of these selenoproteins are a consequence of modifications occurring in mRNA translation or from increased degradation due to growing instability [2]. The influence of Se on the proper functioning of the organism is described later in this section.

#### 3.1.1. Glutathione Peroxidases

Glutathione peroxidases (GPxs) are among the first discovered and described selenoproteins. In this family, eight isoforms (Gpx1–Gpx8) found in mammals are distinguished. These isoforms are characterized by a different structure, localization in tissues, or substrate specificity. Additionally, this family is divided into two subgroups—Se-dependent and Se-independent enzymes. In the Se-dependent enzymes (GPx1–4 and GPx6), the selenocysteine residue is at the active site of the enzyme, whereas in the Se-independent enzymes (GPx5, GPx7, and GPx8), instead of selenocysteine, there is cysteine [2,85,86]. Their main function is to defend from cellular oxidative stress (reduction of peroxides, reactive oxygen, and nitrogen species) [80], whereas more detailed functions of particular Gpx are presented in Figure 3.

There is a strong relationship between serum Se levels and cell redox balance—consumption of this micronutrient has a direct proportional impact on the activity of Se-dependent GPxs [82]. For the reduction of both H_2_O_2_ and organic peroxides (ROOH), GSH is also required in addition to GPx. The reaction cycle produces hydroxyl derivatives of peroxides, i.e., water or their corresponding alcohols, and oxidized glutathione (GSSG) is reduced to GSH by NADPH/H^+^ and GR. The whole process of peroxides and GSSG reduction can be written as follows [87]:(1)$$\mathrm{ROOH}+2\;\mathrm{GSH}\;\overset{{\mathrm{GP}}_{x}}{\to }\;\mathrm{ROH}+{\;H}_{2}O+\mathrm{GSSG}$$
$$\mathrm{GSSG}+\mathrm{NADPH}+{\;H}^{+}\overset{\mathrm{GR}}{\to }2\;\mathrm{GSH}+{\;\mathrm{NADP}}^{+}$$

Redox imbalance and oxidative stress occur when there is an excessive generation of ROS and/or depletion of antioxidants in the cell. GSH belonging to the antioxidant system plays a very important role as it donates reducing equivalents (H^+^) in reactions catalyzed by GPxs. As a result of the transfer of H^+^ to ROS, these molecules become more stable and less toxic (above reaction). If a substance with a pro-oxidant effect on thiols, such as selenites, is present in the cell environment, GSH depletion occurs. This is caused by the oxidation of the -SH groups of the GSH molecule, so that the donation of reducing equivalents is impossible and oxidative stress occurs [90,91]. A very important role of GSH is described for GPx4, as this enzyme protects cells from lipid peroxidation. The depletion of GSH in the cell and thus the lack of reducing equivalents causes the formation of lipid peroxides and their accumulation in the cell membrane. As a result of this phenomenon, non-apoptotic iron-dependent cell death called ferroptosis is induced [83,91].

In contrast, Se-independent GPxs, which have cysteine instead of selenocysteine in the active site of the enzyme, have comparatively low peroxidase activity, therefore their role as a potential GPx is debatable [85,92].

#### 3.1.2. Thioredoxin Reductases

Thioredoxin reductase (TrxR) is a Se-dependent flavoenzyme and is responsible for the reduction of oxidized thioredoxin (Trx) [93,94]. This enzyme, as a flavoprotein, is likely susceptible to low concentrations of vitamin B_2_ (riboflavin), as is glutathione reductase (GR) [95]. Together with thioredoxin and NADPH/H^+^, it represents the so-called thioredoxin-thioredoxin reductase system (electrons from NADPH are transferred by TrxR to Trx) [93,94,96]. In 2017, Yang et al. [97] proved that Trx regulates the concentration of selenoproteins (including Gpx1–4 and TrxR) in chickens by affecting the gene expression of these proteins. In turn, the activity of TrxR affects the functioning of Trx [98], which indicates that three components are responsible for regulating the expression of selenoproteins: Se, TrxR, and thioredoxin [99,100]. As strong electrons donors, thioredoxins are reducers for enzymes, e.g., ribonucleotide reductase during the synthesis of deoxyribonucleotides or thioredoxin peroxidase, which is part of the antioxidant system [101]. The TrxRs family includes three thioredoxin reductases, TrxR1, TrxR2, and TrxR3 [94,96], whose functions are described in Figure 4.

Furthermore, TrxR is responsible for the reduction of low molecular weight substrates such as GSSG [106], lipid peroxides [107], H_2_O_2_ [107], vitamin K [108], and L-dehydroascorbic acid [93].

#### 3.1.3. Thyroid Gland and Iodothyronine Deiodinases

Compared to other parts of the human body, the thyroid gland is characterized by a significant amount of Se per mass unit [7,109]. This micronutrient plays an important role in the proper functioning of the thyroid gland. It can be found in iodothyronine deiodinases (DIOs), which belong to oxidoreductases. This family includes three isoforms of DIOs—DIO 1, 2, and 3, which are responsible for regulating the biological activity of thyroid hormones [109,110,111,112,113]. As shown in Figure 2, they are present in both the fetus and mature organism, controlling the correct course of cell metabolism, growth, and development [112,114,115]. Detailed functions of individual DIOs are given in Figure 5. The deiodation process takes place in peripheral tissues, especially in the skeletal muscles, liver, or kidneys, and may be disturbed by too low Se levels in plasma [111,112]. This leads to iodine release disruption, affecting abnormal thyroid function, which plays an important function in the regulation of thermogenesis and lipid metabolism [7,112]. Additionally, a deficiency of both of these elements may lead to impairment of brain function in adults [116]. For this reason, the important role of both Se and iodine in the proper functioning of this gland as well as the nervous system can be highlighted [7,109,113].

#### 3.1.4. Selenophosphate Synthetase 2

Selenophosphate synthetase 2 (SPS2), much like other representatives described in this section of this review, belongs to the group of selenoproteins. This enzyme is important for the existence and synthesis of selenoproteins [120]. SPS2 is responsible for the formation of selenophosphate in the selenide, adenosine triphosphate (ATP), and H_2_O reaction, in which it is a catalyst [120,121,122]. The product of this reaction (selenophosphate) plays the role of an active Se donor [122], which is crucial during the biosynthesis of Sec. As mentioned earlier, this essential amino acid is part of selenoproteins [121].

#### 3.1.5. Other Selenoproteins

Besides enzymes from the GPxs, TrxRs, and DIOs families, as well as SPS2, other selenoproteins also have Se in the form of Sec residues in their structure. They take important participation in many life processes and are responsible for maintaining their balance in the human body. The best known and characterized selenoprotein, apart from GPx and DIO, is Sepp1, the first evidence of which appeared in the early 1970s [123,124]. SelP mainly occurs in the liver, testes, and brain (Figure 2) and differs from other selenoproteins in that it has 10 residues of Sec in its structure (nine Sec residues included in the short C-terminal domain and one Sec residue in the N-terminal catalytic domain) compared to GPxs, which has only one residue of Sec [125,126]. This group also includes selenoprotein R (SelR), also known as methionine sulfoxide reductase B1 (MsrB1), which is the only one in the Msr antioxidant family that contains Se—the other members are MsrA, MsrB2, and MsrB3 and have cysteine instead of selenocysteine [127]. SelR and other human selenoproteins and their functions in the body are shown in Figure 6.

### 3.2. Effects of Se on Immune Response

We can observe that Se, which is a component of many selenoproteins, performs many different functions in the body. This microelement is also involved in the regulation of the immune system and immune response. It increases the phagocytic activity of macrophages (increase in their cytotoxicity) and stimulates the production of antibodies (IgG and IgM classes) by B lymphocytes. In a study carried out in 2010 by Hoffmann et al. [134] on mice fed low, medium, and high doses of Se in food for eight weeks, it was shown that an intake of 1.0 mg/kg (high dose) of Se increased the proliferation of T_h_ lymphocytes (helper) with a shift in the T_h_1/T_h_2 balance towards the T_h_1 subclass. The rise in the T_h_1 population increased CD40 ligand levels and the production of interferon γ (IFN-γ), which is responsible for stimulating antibody production and strongly activating macrophages. Moreover, an increase in interleukin 2 (IL-2) expression and greater activity of interleukin 2 (IL-2R) receptors with high affinity were observed.

Furthermore, Se affects the activity of natural killer (NK) cells and T_c_ lymphocytes (cytotoxic). In a clinical study conducted by Kiremidjian-Schumacher et al. [54] on people who supplemented Se (as sodium selenite) at a dose of 200 µg/day for eight weeks, an increase in NK cell activity and cytotoxicity of T_c_ lymphocytes was observed, which the researchers associated with an increase in IL-2R receptor expression. Another study by the same research team (1996) [135] showed increased NK cytolysis in C57B1/6J male mice fed food containing 2.0 mg/kg Se for eight weeks. This study also showed a significantly higher amount of IL-2R/cells, which together with a concentration of 0.1 µM Se as sodium selenite led to the transformation of resting NK cell populations into plastic-adherent lymphokine-activated killer (A-LAK) cells and an increase in their proliferation, development, and lytic activity. Moreover, in another study, in cattle (Nellore bulls) fed with feed enriched with 2.5 mg Se and 500 IU vitamin E, an improvement in the cytolytic function of NK cells was observed [136].

Se may also affect the immune system during viral and bacterial infections [137]. A systematic review of clinical trials carried out by Muzembo et al. in 2019 [69] shows that this trace element may delay the progression of HIV to full-blown AIDS, but it does not inhibit/reduce the viral load of the virus. Besides, in the case of hepatitis B and C viruses, this microelement exhibits hepatoprotective properties [138] and also inhibits the progression of influenza and polio infection [139].

It is now assumed that Se and its levels in serum may have an impact on the number of SARS-CoV-2 infections and the course of COVID-19 [139,140], which is associated with a reduced expression of selenoproteins [140]. It is postulated that SARS-CoV-2, like other viruses causing respiratory infections, may affect the cell redox balance as well as being responsible for ER stress and the development of an inflammatory process against which Se and selenoproteins protect [141]. In this case, GPx1, which interacts with the viral M^pro^ (Nsp5) protease required for replication, is very important [142]. In addition, too low Se concentrations shift the balance between prostaglandin I2 (PGI_2_) and thromboxane A2 (TXA2) towards TXA2 in rats [143], resulting in increased blood coagulation and vasoconstriction [144]. These findings may explain the development of venous thromboembolism in COVID-19 patients with suboptimal Se levels [144]. In a study carried out by Moghaddam et al. (*p* < 0.001) [145], it was observed that survivors of COVID-19 had significantly higher Se status, but this cannot be considered the cause of the disease and the severity of its course due to the observational nature of the study. Meanwhile, results from a study by Majeed et al. [146] showed that healthy participants had higher serum Se levels (79.1 + 10.9 ng/mL) compared to people suffering from COVID-19 (69.2 + 8.7 ng/mL). Very important results are shown in an analysis by Zhang et al. [147], who assessed the correlation between the COVID-19 recovery rate and hair Se concentration. The analysis included multiple regions of China with different Se content in the soil. Higher recovery rates were observed in areas with high soil Se content—Enshi (Hubei Province, South China) compared to Heilongjiang Province (northeastern China, a place where the KD was discovered), which is characterized by low Se status. In addition, higher mortality rates were observed in areas with existing Se deficiency. Referring to these data, it can be concluded that the probability of recovery from COVID-19 was dependent on Se concentration and increased with its levels. For this reason, it is recommended to supplement Se and other micronutrients, i.e., zinc and vitamins D, E, and C, which affect the body’s immunity [139,148,149].

Furthermore, among recent reports, it seems interesting that sodium selenite (SS), which is a simple inorganic compound, can inhibit the entry of the SARS-CoV-2 virus into healthy cells. Its probable mechanism of action involves a pro-oxidative effect on thiol groups present in the protein disulfide isomerase (PDI) found in the virus, blocking its ability to cross cell membranes and abrogate infectivity [150]. Due to these properties, SS has the potential to be used in the eradication of the SARS-CoV-2 virus, but more research is needed on this topic, especially regarding the safety and doses used in humans that show this desirable activity.

### 3.3. Effects of Se on the Reproductive System and Other Body Functions

Much has been said about the role of Se in the fertility and reproduction of humans and animals. A deficiency of this trace element results in poorer quality sperm. Morbat et al. [151] observed that in 12 infertile men, who were given 50 µg Se/day for three months, sperm quality had improved in terms of number, viability, motility, and spermatozoa morphology. The same conclusions were reached in randomized controlled clinical trials (RCTs) performed in Scotland, Tunisia, and Iran. This is due to an Se deficiency and resulting low GPx4 status, leading to a redox imbalance in the cells and the formation of large amounts of ROS (reactive oxygen species), which damage the spermatozoa [65]. Additionally, Se is essential for the biosynthesis of testosterone, which also affects the quality of sperm, and this relationship is directly proportional [6,152].

In the case of women, Se is very important for embryo implantation and pregnancy maintenance. Observational studies in the United Kingdom and Turkey have shown that women who had miscarried in the first trimester of pregnancy had a significantly lower serum Se status. It is suspected that the loss of pregnancy was due to too low Gpx and too little antioxidant protection of DNA and biomembranes [6,65]. Additionally, Se deficiency may be associated with pregnancy complications or may be harmful to the fetus. These adverse effects manifest themselves in the form of damage to the immune and/or nervous system of the developing fetus, and too low Se level at the beginning of pregnancy can predict low birth weight of the baby [6]. Unfortunately, there is not much research on women and their fertility, and this data is very sparse.

The last important function of Se is its role in the aging process of the body, including the skin. This element delays skin aging [153,154] and protects this organ from photoaging caused by UVA radiation [154]. Its increased concentration in the body affects higher levels of selenoproteins, which have antioxidant and anti-inflammatory effects, preventing the aging process [9,155]. Additionally, Se prevents tissue flabbiness, improving their flexibility, and also reducing the intensity of menopause symptoms [1].

## 4. Selenium and Cancer Chemoprevention

### 4.1. Clinical Trials and Other Types Of Studies on the Chemopreventive Effects of Selenium

The problem of cancer affects the whole world. Year after year, more and more new cancer cases are reported. The most common types of cancer in the population are lung, breast, colorectal, and prostate cancer, of which lung cancer has caused the most deaths [10,156]. For many years, just these types of cancer were the main subject of studies on the relationship between Se intake and the frequency of their incidence [4,5,45,46,48,49,51,81,128]. Many studies, both on animals and humans, have shown that low Se status (resulting from its low intake) correlates with a higher risk of developing cancer [50,55]. In these studies, an anticancer effect of Se, i.e., an inverse relationship between its consumption and cancer incidence, was observed when this element was taken in supranutritional doses in a range of 250–300 µg/day [1,4].

Despite this, some studies contradict this fact [45,81]. Recent and current clinical trials on the chemopreventive effect of Se in the most common cancers are shown in Table 3. Because of these inconsistencies, the chemopreventive effect of Se is still ambiguous and requires further, more detailed research to clarify this relationship. Of course, the fact that many different Se compounds have been used in studies cannot be ignored. It is also worth noting that the effect, and therefore the outcome, of the research may depend on the chemical form of the Se compound (inorganic compound, e.g., sodium selenite or organic compound, e.g., SeMet), the taken dose, the bioavailability of this element, the baseline selenium status of the study population, the type of cancer, and even the stage of cancer lesions [1,4,157].

Precisely due to the different research results in this area, many authors have conducted meta-analyses, meta-regressions, and systematic reviews that summarize the results of all the studies on the impact of Se on the risk of developing individual cancers. In the case of breast cancer, the most frequently diagnosed cancer among women, there are three meta-analyses available describing this association. Vincenti et al. [171], when they were performing their meta-analysis, included eight studies and did not confirm correlation of Se supplementation with reduced breast cancer risk. However, Cai et al. [45], who summarized 14 studies, and Babaknejad et al. [47], who analyzed 16 articles, concluded that high exposure and high serum Se concentration significantly reduced the risk of breast cancer development, but its toenail concentration did not prove this relationship significantly. In the case of toenail, the content of this element is measured by neutron activation analysis (NAA), inductively coupled plasma spectrometry (ICP-MS, ICP-OES, ICP-AES), or atomic absorption spectrometry (AAS) after preliminary preparation of the sample. These methods are characterized by detection of the element only, e.g., Se and the result is usually given as µg element/g of the toenail. Importantly, the content of this micronutrient in the toenail is a biomarker of long-term Se exposure and thus can be successfully used in retrospective studies. In this regard, the difference observed between high serum Se concentration and its low content in toenail may indicate that supplementation was short-term (serum Se levels increase rapidly, but slowly in toenail). Therefore, these two parameters should not be compared if Se supplementation in the study was not long-term, as this may falsify the conclusions [172].

Among sex-specific neoplasms, prostate cancer is one of the most commonly diagnosed (191,930 cases in 2020) [10]. In a randomized, double-blind, placebo-controlled trial called the Nutritional Prevention of Cancer (NPC), which started in the 1980s, lasted almost seven years (until 1991), and involved 1312 patients with a history of non-melanoma skin cancer, an effect on the primary endpoint was not observed. However, after analyzing the secondary endpoints, Clark et al. [173] showed that 200 µg/day of Se supplementation in the form of Se-yeast reduced the risk of prostate, colorectal, and lung cancer. Thanks to these conclusions, many authors have undertaken further and broader research on the chemopreventive effects of this element against prostate cancer. One of the most anticipated results of the research in this area was a large-scale study called the Selenium and Vitamin E Cancer Prevention Trial (SELECT), launched in 2001. Almost 35,600 men were included in this study, who were taking Se + vitamin E, Se placebo + vitamin E, Se + vitamin E placebo, or placebo + placebo (200 µg/day SeMet, 400 IU vitamin E), depending on the arm of the trial. However, the study was terminated prematurely due to the lack of a visible chemopreventive effect of these antioxidants against prostate cancer. Besides, in the case of the vitamin E arm, an increased risk of prostate cancer was reported. This decision was made by the independent Data and Safety Monitoring Committee (DSMC), the Southwest Oncology Group (SWOG), and the National Cancer Institute (NCI) [174]. However, the results of the SELECT trial are very often misinterpreted. Firstly, Se-enriched yeast was used in the NPC while SeMet was supplemented in the SELECT. The fact is that, despite using similar doses of Se, Se-enriched yeast can have variable amounts of this amino acid as well as other selenoamino acids, e.g., MSC [175,176]. Secondly, the effectiveness of supplemented Se is mainly dependent on its baseline blood levels. The lack of reduction in cancer cases in the SELECT study does not indicate the lack of a chemopreventive effect of Se but rather confirms the results of the NPC study. This is because, in the NPC, only participants with Se levels <114 ng/mL showed a significant reduction in prostate cancer risk, whereas in the SELECT trial participants had higher baseline levels of this trace element (>135 ng/mL). This difference was related to the place of residence of the individual participants. Participants in the NPC were from regions with low Se status in the environment (eastern coast of the USA), whereas participants in the SELECT lived in Canada and the USA, where the content of this element in the soil is higher. This suggests that even the lowest baseline of Se levels in the SELECT trial participants may be higher than those required for a chemopreventive effect, and therefore noticeable effects may be observed only in groups deficient in this micronutrient [177,178]. In addition, in the search for clear and unambiguous results, many authors have conducted meta-analyses and systematic reviews combining all studies on Se and its effect on prostate cancer development in men. Each of them showed that high concentrations of Se in serum, plasma, or toenails were associated with a lower risk of prostate cancer [44,179,180,181,182].

The second most frequent cancer among both sexes is lung cancer, which caused the most deaths [10]. In meta-analyses and systematic reviews concerning the development of lung cancer compared to plasma/serum/toenail Se levels, the authors have shown a positive chemopreventive effect of Se on this type of cancer [45,46,183,184]. Among the most commonly reported cancers of both sexes, colorectal cancer ranks third place. It was detected in 147,950 people in the United States only in 2020, of whom 69,650 (8%) were women and 78,300 (9%) were men [10]. A meta-analysis of the Women’s Health Initiative Observational Study and 14 previous studies (clinical trials and observational studies) revealed that the relationship between Se concentration and colorectal cancer is inverse and only relevant for men, while for women it was not demonstrated [185]. In the Cochrane Systematic Review from 2018 [171], an analysis of databases of RCTs with a total of 19,009 participants, did not prove the effect of Se on colorectal cancer risk. On the other hand, during an analysis of 15 observational cohort studies (covering almost 2,400,000 patients), the authors of this review showed that high Se exposure reduced the risk of colorectal cancer more among men than women. However, in their opinion, these studies had major deficiencies/mistakes and were of poor quality, thus these results cannot be taken as clear and reliable. In their paper, Cai et al. [45] contradicted the protective role of Se in the development of colorectal cancer, whereas Ou et al. [186] proved it only in the case of pre-malignant lesions, i.e., colorectal adenomas.

### 4.2. Molecular Mechanisms of the Chemopreventive Activity of Selenium

As has already been proven, Se plays a major role in the prevention of various cancers. Nevertheless, so far the specific molecular and genetic mechanisms of its action have not been fully known. This micronutrient has different multilevel effects and it is difficult to determine the main one. As oxidative stress is one of the leading factors in the neoplastic transformation of cells, it is recognized that the main mechanism responsible for the protective properties of this trace element is antioxidative cell protection. The antioxidant activity results in a reduction in the production of reactive oxygen species (ROS) and a decrease in DNA and cell membrane damage caused by these free radicals [4,55]. It is worth emphasizing the fact that the chemical form of the ingested Se compound is an important factor that determines its effectiveness in preventing DNA damage in cells. As it turns out, SeMet exhibits lower protection against DNA damage than Sec [187], but, as is well known, Se maintains a redox balance in cells through the selenoproteins that contain it in their active center in the form of Sec [4,51,55]. The selenoproteins whose expression or polymorphisms could be linked to antioxidant protection or cancer development mainly include Gpxs, TrxRs, and Sepp1 [55,81,82]. An increase in Gpx1 and TrxR activity, which was associated with Se supplementation in the form of sodium selenite (30 nM) and SeMet (10 µM) at non-cytotoxic doses, was observed as protection against UVA, or hydrogen peroxide, induced genotoxicity. In addition, an increase in the DNA repair capacity was found [132,188].

Polymorphisms concerning selenoprotein genes are also an important issue. For example, polymorphisms within the TrxR2 gene may be associated with an increased risk of developing aggressive prostate cancer, while differences in SelP and TrxR2 genes may have an impact on Se concentrations in plasma. Moreover, an influence of selenoprotein polymorphisms on the risk of breast (Gpx1, SelF, SelP) or colorectal (Gpx4, TrxR1, and TrxR2, SelP, SelF, and SelS) cancer was also found, which could indicate their important role in the etiology of these types of cancer [81].

In the case of the Gpxs family, studies have described their association with the incidence of cancer. For instance, Gpx1 is responsible for preventing DNA mutation, and the disappearance of the heterozygous character of the Gpx1 gene (anti-oncogenic effect) commonly occurs in the case of breast, colon, head, or neck cancer. Mutations in the Gpx1 gene increased the risk of developing cancer, whereas overexpression of normal selenoprotein showed a protective role [82,87]. Besides, it was proven that a reduction in Gpx4 levels occurs in the case of breast and pancreatic cancer, so this selenoprotein could be considered to have the same suppressive effect on tumors as Gpx1 [87]; for Gpx2, its protective or pro-oncogenic role remains unresolved [132].

Except for Se’s antioxidant effect, it also increases the immune response of the body and reduces inflammation [5,51,54,79]. This is possible thanks to the stimulation of cytotoxic T lymphocytes (T_c_) and macrophages, NK cell proliferation, and differentiation into cytotoxic effector cells and an increase of their activity, which has been thoroughly described earlier in this review. Among the other chemopreventive mechanisms of Se, DNA stabilization, the growth of p53 protein expression, suppression of protein kinase C (PKC), interruption of the cell cycle, directing cells to the apoptosis pathway, changes in methylation of the regulatory part of genes in DNA, histone modifications, interference with one-carbon metabolism, modulation of cell multiplication, suppression of angiogenesis, induction of enzymes participating in the second phase of xenobiotics biotransformation, prevention of toxic effects of heavy metals on cells, and regulation of the expression of estrogens as well as androgen hormones receptors stand out [4,51,189]. The cell cycle is arrested by Se in the G1 phase in all cell types; moreover, the expression of genes, i.e., cyclin A and D1, cell division cycle 25 A protein (Cdc25A), cyclin-dependent kinase 4 (CDK4), proliferating cell nuclear antigen (PCNA), and transcription factor 2 (E2F) are inhibited. At the same time, the gene expression of Arf protein (p19), CDK-interacting protein 1 (cyclin-dependent kinase inhibitor 1, p21), p53 protein, glutathione S-transferase (GST), superoxide dismutase (SOD), NAD(P)H dehydrogenase [quinone] 1 (NQO1), growth arrest and DNA-damage-inducible protein (GADD153), as well as several caspases, are stimulated [190]. As is known, an increase in p53 protein expression leads to an arrest of the cell-cycle and directing the cell to the apoptotic pathway, so this gene is important in controlling DNA repair and regulating the life-cycle of cells. Moreover, its activity is regulated by Se levels in plasma. In the case of cancer development, mutations in the gene encoding this protein have been reported [191]. Additionally, it was proven that Se is also important in inhibiting the development of cancer metastases, as it blocks the synthesis of the osteoponin responsible for this process [192]. A summary of the possible mechanisms of cancer prevention by Se are presented in Figure 7.

It is also worth noting that Se can exert its chemopreventive effects not only through selenoproteins but also with the direct involvement of Se metabolites. One such metabolite is methylselenol, whose chemopreventive as well as anticancer effects have been demonstrated in many publications [83,193,194]. This compound can be generated from both inorganic (selenites, selenides) and organic compounds (natural: SeMet, MSC, and synthetic origin, e.g., methylselenic acid, selenoesters) [83]. In the case of SeMet and MSC, this compound is produced via the enzyme-liase activity (γ-liase in the case of SeMet, β-lyase—MSC) [58,83,195]. In contrast, inorganic compounds are first metabolized to selenide, which then undergoes methylation to form methylselenol [58,83,196,197]. It was observed that this metabolite leads to cell cycle arrest in the G1 phase [198,199,200] and directs cancer cells to the apoptotic pathway [193,194,196,201,202]. Among other effects of this Se metabolite, suppression (inactivation) of PKC [193] and NF-κB [203], inhibition of cell growth [193,200], regulation of gene expression [200], an increase in p53 protein expression, and stimulation of DNA repair [204] were reported. Additionally, it was shown that this compound suppresses angiogenesis and regulates the expression of sex hormone receptors [194,197,199,205]. Because of the above, it can be concluded that the chemopreventive effect of Se is associated with both the selenoproteins in which it is incorporated and the metabolites containing this element. During the selection of Se compound for dietary intervention, it is worth being guided by the properties resulting from its chemical form, as well as other potential benefits associated with the mechanism of metabolites activity. However, it should be remembered that, in humans, γ-liase has a very low activity, therefore the metabolism of SeMet to methylselenol is negligible [195].

## 5. Conclusions

Selenium (Se) is an essential micronutrient that is responsible for the proper functioning of the body, which is why its adequate daily intake is so important. Its impact on redox balance, hormonal management, the immune system, and fertility mainly manifests through selenoproteins such as GPxs, TrxRs, DIOs, or others, e.g., SelP, into which it is incorporated via the metabolism of Se-containing compounds ingested with food. However, it should be remembered that this trace element is characterized by a narrow range between proper levels and deficiency/overdose. In dietary doses, it performs an antioxidant activity after incorporation into selenoproteins, while in supranutritional doses, it is a prooxidant, exhibiting toxic effects on the body by ROS.

The worldwide statistics concerning cancer show fearful numbers of patients with this disease. For this reason, possible prophylactic measures are being sought to prevent cancer development and thus reduce the number of new cases and deaths. Se may be one such preventive agent. The main mechanisms by which Se manifests its chemopreventive activity include DNA repair/stabilization and inhibition of cancer progression. Although most meta-analyses, meta-regressions, and systematic reviews conclude that Se can exert a chemopreventive effect, it would be appropriate to conduct further extensive research in this field. This could provide data that would ultimately help to determine whether Se exhibits chemopreventive activity and the exact mechanism of this phenomenon.

## Figures and Tables

**Figure 1 nutrients-13-01649-f001:**
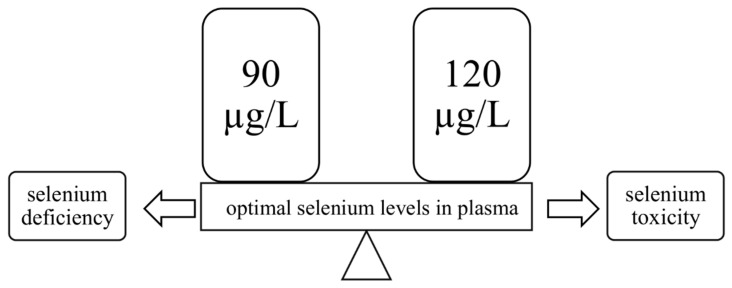
The range of normal (optimal) plasma selenium levels. Values above and below this range indicate selenium toxicity or deficiency, respectively.

**Figure 2 nutrients-13-01649-f002:**
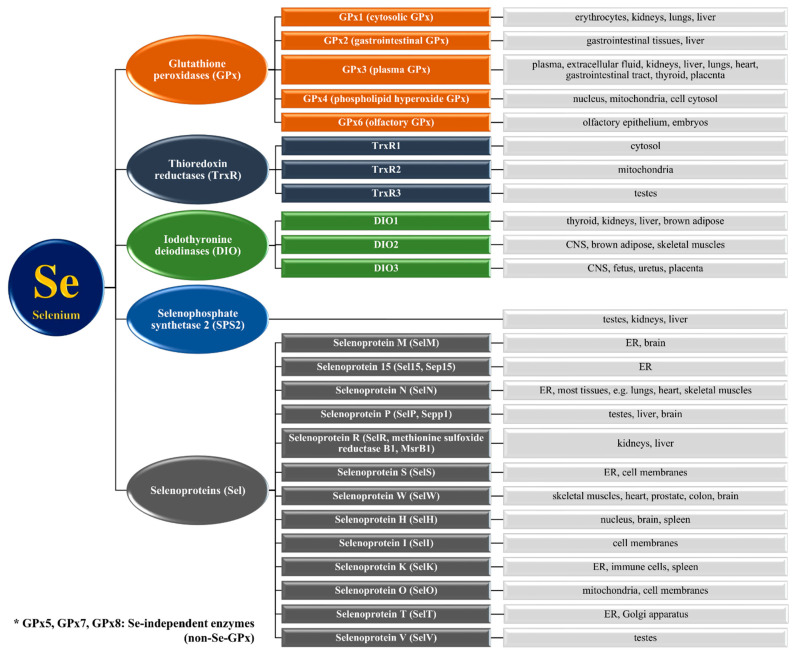
Selenoprotein families and their localization in the human body [2,82,85,86]. ER—endoplasmic reticulum, CNS—central nervous system.

**Figure 3 nutrients-13-01649-f003:**
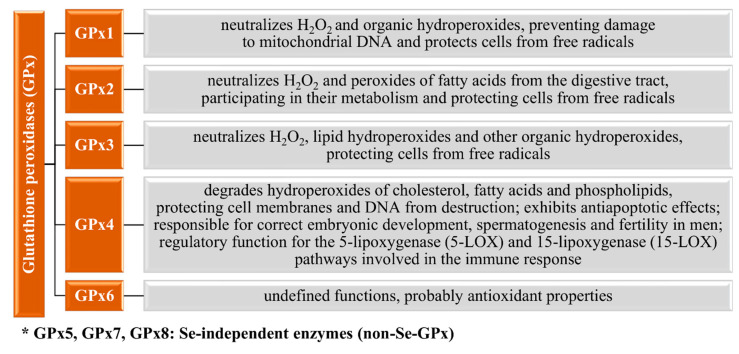
Selenoproteins of the glutathione peroxidases (GPx) family and their functions [85,87,88,89].

**Figure 4 nutrients-13-01649-f004:**
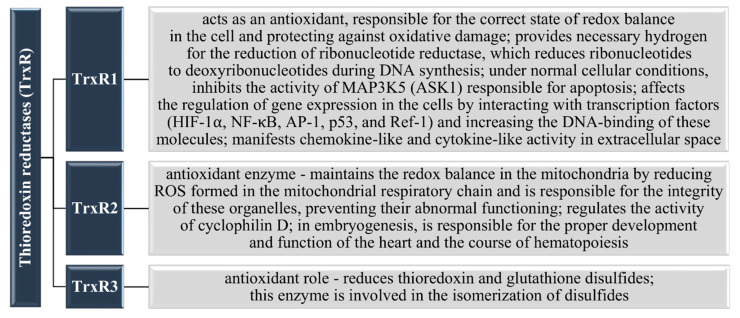
Selenoproteins of the thioredoxin reductases (TrxR) family and their functions [102,103,104,105].

**Figure 5 nutrients-13-01649-f005:**
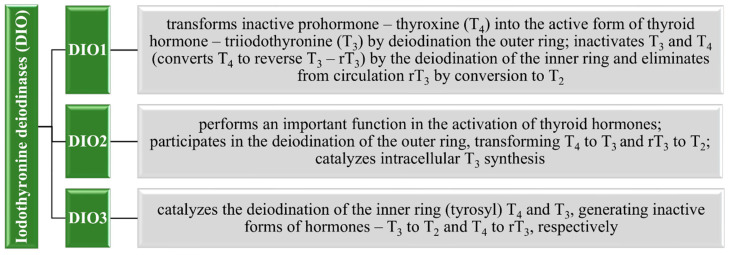
Selenoproteins of the iodothyronine deiodinases (DIO) family and their functions [114,115,117,118,119].

**Figure 6 nutrients-13-01649-f006:**
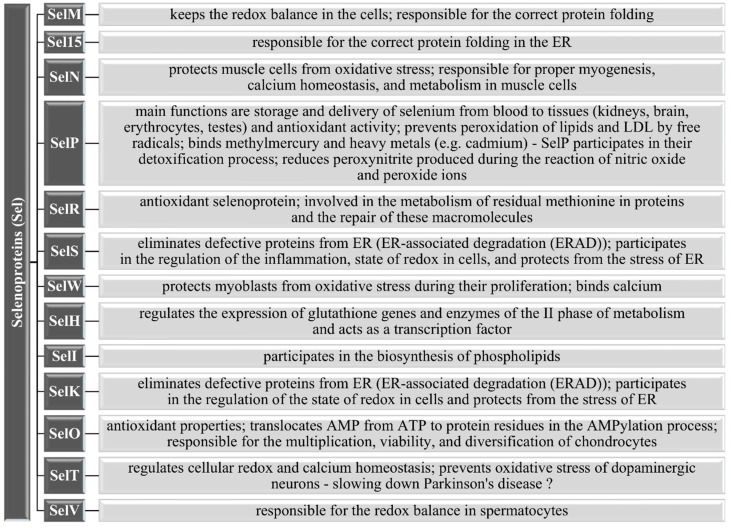
Other selenoproteins and their functions [8,55,82,86,128,129,130,131,132,133].

**Figure 7 nutrients-13-01649-f007:**
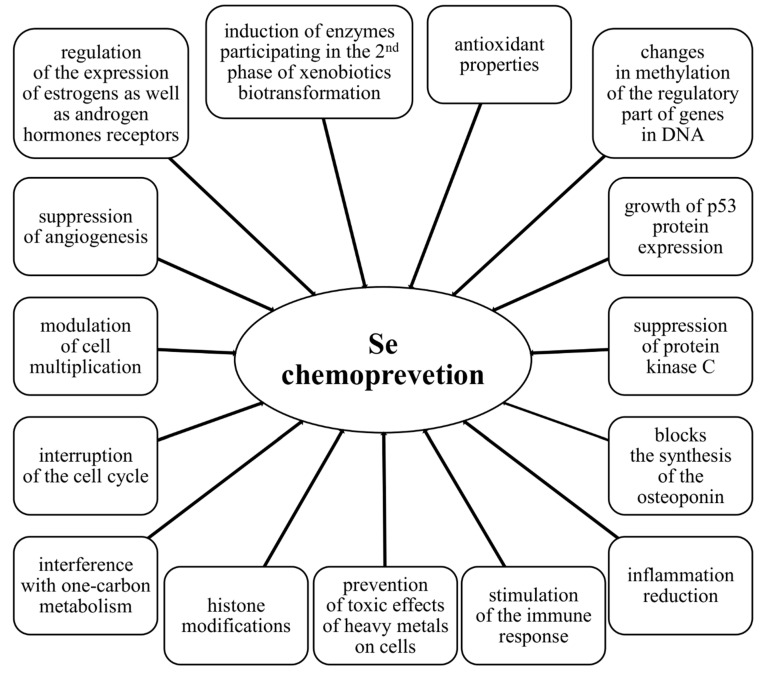
Possible mechanisms of cancer chemoprevention by selenium.

**Table 1 nutrients-13-01649-t001:** Examples of some food products and their selenium content.

Food	Typical Se Content (µg/g Fresh Weight)	References
Se-yeast	3000	[15]
Broccoli (Se-enriched)	62.3	[16]
Brazil nuts	0.85–6.86	[17]
Shellfish	0.36–1.33	[18]
Chicken	0.57	[19]
Beef	0.35–0.47	[20]
Lamb	0.4	[21]
Salmon	0.21–0.27	[22]
Eggs	0.17	[23]
Milk products	0.1–0.55	[17]

**Table 2 nutrients-13-01649-t002:** Recommended daily intakes of selenium for individuals in different age groups and genders. The Adequate Intake (AI) is equivalent to the mean selenium intake of healthy, breastfed infants from birth to 12 months of age, as established by the Food and Nutrition Board (FNB) [13].

Age Group	Male (µg/d)	Female (µg/d)	Pregnancy (µg/d)	Lactation (µg/d)
Birth to 6 mths	15 *	15 *		
7 to 12 mths	20 *	20 *		
9 to 13 yrs	40	40		
14 to 18 yrs	55	55	60	70
19 to 30 yrs	55	55	60	70
31 to 50 yrs	55	55	60	70
50 to 71 yrs	55	55		
Above 71 yrs	55	55		

* Adequate Intake (AI).

**Table 3 nutrients-13-01649-t003:** Recent and current clinical trials on selenium chemoprevention in breast, colorectal, lung, and prostate cancer. This table was compiled from the information available at https://www.clinicaltrials.gov/ (accessed on 25 October 2020).

NCT Identification Number	Study Title	Clinical Trial Status	Type of Cancer	Study Design	References
NCT00555386	Soy, selenium, and breast cancer risk	Not applicable ^b^	Breast Cancer	Randomized, double-blind (participant, investigator), placebo-controlled, parallel assignment	[158]
NCT04014283	Prevention of female cancers by optimization of selenium levels in the organism (SELINA)	Not applicable ^c,d^	Breast Cancer	Randomized, single-blind (investigator), placebo-controlled, parallel assignment	[159]
NCT01611038	Chemoprevention of breast and prostate cancers in shift workers by dietary methylselenocysteine: effects on circadian rhythm and estrogen receptor-B cycling	Not applicable ^b^	Breast and Prostate Cancer	Randomized, triple-blind (participant, investigator, outcomes assessor), placebo-controlled, parallel assignment	[160]
NCT01211561	Colon cancer prevention using selenium	Unknown ^b,^*	Colorectal Cancer	Randomized, double-blind (participant, investigator), placebo-controlled, single group assignment	[161]
NCT00706121	S0000D: Effect of vitamin E and/or selenium on colorectal polyps in men enrolled on SELECT trial SWOG-S0000 (ACP)	Phase 3 completed ^b,+^	Colorectal Cancer	Randomized, quadruple-blind (participant, care provider, investigator, outcomes assessor), placebo-controlled, parallel assignment	[162]
NCT01437826	Antioxidant supplement and reduction of metachronous adenomas of the large bowel: a double blind randomized trial	Phase 3 terminated ^a^	Colorectal Cancer	Randomized, double-blind (participant, investigator), placebo-controlled, parallel assignment	[163]
NCT00008385	Selenium in preventing tumor growth in patients with previously resected stage I non-small cell lung cancer	Phase 3 completed ^a,+^	Lung Cancer	Randomized, triple-blind (participant, care provider, investigator), placebo-controlled, parallel assignment	[164]
NCT00978718	Selenium in preventing prostate cancer	Phase 3 Completed ^b^	Prostate Cancer	Randomized, double-blind, placebo-controlled	[165]
NCT00446901	Selenium and prostate cancer: clinical trial on availability to prostate tissue and effects on gene expression (SePros)	Not applicable ^b^	Prostate Cancer	Randomized, quadruple-blind (participant, care provider, investigator, outcomes assessor), placebo-controlled, parallel assignment	[166]
NCT00006392	S0000: Selenium and vitamin E in preventing prostate cancer (SELECT)	Phase 3 completed ^+^	Prostate, Cancer	Randomized, quadruple-blind (participant, care provider, investigator, outcomes assessor), placebo-controlled, factorial assignment	[167]
NCT00030901	S9917: Selenium in preventing cancer in patients with neoplasia of the prostate	Phase 3 completed ^a,+^	Prostate Cancer	Randomized, triple-blind (participant, care provider, investigator), placebo-controlled, parallel assignment	[168]
NCT00064194	Vitamin E, selenium, and soy protein in preventing cancer in patients with high-grade prostate neoplasia	Phase 3 completed ^b^	Prostate Cancer	Randomized, triple-blind (participant, care provider, investigator), placebo-controlled	[169]
NCT01497431	Se-methyl-seleno-L-cysteine or selenomethionine in preventing prostate cancer in healthy participants	Phase 1 completed ^b^	Prostate Cancer	Randomized, double-blind (participant, investigator), placebo-controlled, parallel assignment	[170]

^a^ study started before the SELECT trial. ^b^ study started during the SELECT trial. ^c^ study started after results of the SELECT trial. ^d^ recruitment to the target population is ongoing. ^+^ results of the study are available on ClinicalTrials.gov. * no verification of status >2 years despite completion date has passed.

## Data Availability

Not applicable.

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
