# Peer review of "Selenium as a Bioactive Micronutrient in the Human Diet and Its Cancer Chemopreventive Activity"

_nutrients, 2021, doi:10.3390/nu13051649_

Round 1

Reviewer 1 Report

Brief summary

The authors outline the main known functions of Selenium in human health, including a brief overview of Se biology and focus on selenoprotein mediated effects. The well cited examples of Se excess and deficiency are discussed. The more novel aspects of the review centre on Se’s effects on immune function and its interactions with viral pathogens. Lastly the review revisits the controversial area of cancer chemoprevention, highlighting a number of mechanisms that might mediate anti-cancer effects in vivo.

Broad comments

Whilst this review does cover material published in a  number of Selenium reviews, I believe it contains a detailed summary of known selenoprotein function and does remind readers of that even after SELECT Selenium may yet have a role in cancer chemoprevention for certain populations.

In the section on dietary intake I think it would be discuss how optimal intake is best assessed ie  Se plasma levels, serum or Plasma SEPP1 or GPX3 activity?? Prof Rayman in her 2012 Lancet review refers to plasma levels 120 to 150mg/L or so for minimal effects on mortality  (NHANES population review).

Under Selenium Overdose, you could mention that even lower doses around of 300mg/ day of Selenium as selenized yeast can have some toxicity if taken for long time periods ( see longterm follow up of the Danish PRECISE study)

In the discussion on molecular mechanisms of chemoprevention, should mention Clement IPs work around methylselenol and pre-malignant lesions in mammary glands, which suggests some direct effects of Se metabolites rather than specific Selenoprotein mediated functions which brings into focus the importance of the choice of Selenium compound or dietary intervention to use.

Specific comments.

LINE 47 “may” rather than “can” probably  a matter of opinion though.

LINE 149 would be useful for the reader if risk could be quantified/ estimated.

LINE 204 “ activity”rather than activeness

LINE 378-382. Worth mentioning baseline Selenium status of study population too.

Table 3: some the studies such as Bonelli  et al commenced in 1980s and many of the study are more than 10 years old,  so an alternative to “recent” should be used.

I think the table does illustrate there was clinical activity in this space, but would be worth noting if available how many of these studies were shut down after SELECT or failed to recruit to its target population and no results were published or are expected.

LINE 406, clarify meaning, needs rephrasing.

LINE 426 could also mention other Se containing compounds such as gamma-MSC.

LINE 435 ? content in soil/ geochemistry

LINE 461. Alternative to mild cancers,  eg. pre-malignant lesions

Figure 7, rephrase epigenetic mechanism.

Author Response

RESPONSE TO REVIEWER #1:

Thank you for your review of our manuscript entitled “Selenium as a bioactive micronutrient in the human diet and its cancer chemopreventive activity” for publication in Nutrients. Overall, we find Reviewer suggestions to be helpful, and constructive, and the corresponding revisions have strengthened the paper in multiple ways. According to your suggestions, we have expanded our manuscript with the content you suggested (the changes made in the manuscript are marked in red).

Broad comments

  • “In the section on dietary intake I think it would be discuss how optimal intake is best assessed ie Se plasma levels, serum or Plasma SEPP1 or GPX3 activity?? Prof Rayman in her 2012 Lancet review refers to plasma levels 120 to 150mg/L or so for minimal effects on mortality (NHANES population review).”

Response: Information on the assessment of Se content has been added to the text. However, we based the information on a more recent publication (Kipp et al. 2015).

  • “Under Selenium Overdose, you could mention that even lower doses around of 300mg/ day of Selenium as selenized yeast can have some toxicity if taken for long time periods ( see longterm follow up of the Danish PRECISE study).”

Response: Information has been added to the text.

  • “In the discussion on molecular mechanisms of chemoprevention, should mention Clement IPs work around methylselenol and pre-malignant lesions in mammary glands, which suggests some direct effects of Se metabolites rather than specific Selenoprotein mediated functions which brings into focus the importance of the choice of Selenium compound or dietary intervention to use.”

Response: Indeed, the chemopreventive mechanism of Se action cannot be associated with selenoproteins alone. A brief discussion of methylselenol based on recent publications has therefore been added.

Specific comments

  • “LINE 47 “may” rather than “can” probably a matter of opinion though”.

Response: it has been changed.

  • “LINE 149 would be useful for the reader if risk could be quantified/ estimated.”

Response: the risk has been estimated.

  • “LINE 204 “ activity”rather than activeness”

Response: it has been changed.

  • “LINE 378-382. Worth mentioning baseline Selenium status of study population too.”

Response: it has been added.

  • “Table 3: some the studies such as Bonelli et al commenced in 1980s and many of the study are more than 10 years old,  so an alternative to “recent” should be used. I think the table does illustrate there was clinical activity in this space, but would be worth noting if available how many of these studies were shut down after SELECT or failed to recruit to its target population and no results were published or are expected.”

Response: The studies before, during and after SELECT were highlighted in the table. Additionally, information of results has been included.

  • “LINE 406, clarify meaning, needs rephrasing.”

Response: it has been corrected.

  • “LINE 426 could also mention other Se containing compounds such as gamma-MSC.”

Response: it has been added.

  • “LINE 435 ? content in soil/ geochemistry”

Response: Indeed, we had in mind to the Se content of the soil. The information has been added.

  • “LINE 461. Alternative to mild cancers, pre-malignant lesions”

Response: it has been changed.

  • “Figure 7, rephrase epigenetic mechanism”

Response: Figure 7 has been corrected.

We hope that our answer will satisfy Reviewer and Editor. We thank you for consideration and await your decision.

Reviewer 2 Report

The Review is well written and comprehensive. All the aspects of Selenium's role in the organism, in physiological and pathological conditions, in particular in cancer, have been extensively analyzed on the basis of Literature data. I have just a few minor observations. 

  • Page 3, also an association with thyroid cancer has been documented in Literature, see “Shen F et al, Biol Trace Elem Res 2015, The Association Between Serum Levels of Selenium, Copper, and Magnesium with Thyroid Cancer: a Meta-analysis” and Kohrle J 2013 J Endocrinol Invest “Pathophysiological relevance of selenium”. 
  • At page 4, in the paragraph regarding “Selenium Overdose” a possible role of Selenium excess producing a pro oxidant effect on thiols like glutathione is presented. Later, at page 7, the role of Selenium in the process of glutathione peroxidase activity, involving also glutathione is given. I think that an explanation of the relationship between the two phenomena should be useful to the reader to avoid confusion. 
  • Page 14: what do you mean with Se “toenail” concentration? This issue should be at least introduced if mentioned.
  • Page 14: the sentence “men are the most commonly diagnosed with prostate cancer” should be written differently, since women cannot indeed.

Author Response

RESPONSE TO REVIEWER #2:

Thank you for your review of our manuscript entitled “Selenium as a bioactive micronutrient in the human diet and its cancer chemopreventive activity” for publication in Nutrients. Overall, we find Reviewer suggestions to be helpful, and constructive, and the corresponding revisions have strengthened the paper in multiple ways. According to your suggestions, we have expanded our manuscript with the content you suggested (the changes made in the manuscript are marked in red).

  • “Page 3, also an association with thyroid cancer has been documented in Literature, see “Shen F et al, Biol Trace Elem Res 2015, The Association Between Serum Levels of Selenium, Copper, and Magnesium with Thyroid Cancer: a Meta-analysis” and Kohrle J 2013 J Endocrinol Invest “Pathophysiological relevance of selenium”.

Response: Thyroid cancer has been added.

  • “At page 4, in the paragraph regarding “Selenium Overdose” a possible role of Selenium excess producing a pro oxidant effect on thiols like glutathione is presented. Later, at page 7, the role of Selenium in the process of glutathione peroxidase activity, involving also glutathione is given. I think that an explanation of the relationship between the two phenomena should be useful to the reader to avoid confusion.”

Response: Information on the effect of oxidative stress on the GSH-GPx system has been added in the text.

  • “Page 14: what do you mean with Se “toenail” concentration? This issue should be at least introduced if mentioned.”

Response: the point about "Se toenail concentration" has been discussed and clarified - an appropriate description is included in the manuscript.

  • “Page 14: the sentence “men are the most commonly diagnosed with prostate cancer” should be written differently, since women cannot indeed.”

Response 4: it has been corrected.

We hope that our answer will satisfy Reviewer and Editor. We thank you for consideration and await your decision.

Reviewer 3 Report

Radomska et al., in their manuscript entitled ‘Selenium as a bioactive micronutrient in the human diet and its cancer chemopreventive activity’ elaborate the role of Se as micronutrient in health and various clinical conditions. The manuscript presents with extensive literature review that also covers the potential role of Se in Covid pathogenesis.  

Major points:

  1. Please include a brief detail of dietary Se absorption and its transport.
  2. Chapter 3.1.1. Glutathione peroxidases. Page 7: The role of GPX4 in ferroptosis (an iron-dependent cell death mechanism) has been completely ignored. Ferroptosis has emerged as an entirely new field during the last decade, as evident from innumerous publications. Please include a discussion of a reasonable length and content.

Minor points:

  1. Section 2.1. Page 2. Line 69: “Richie et al. proved”. It would read better if ‘proved’ is replaced by either ‘observed’ or ‘showed’.
  2. Section 2.1. Page 3. Lines 84-85 & Section 3.1.4. Page 9. Lines 249-251: Two examples are highlighted. Throughout the manuscript, number of sentences are written in this complex manner using multiple which or what. Please simplify the statements for convenient reading.
  3. Section 2.1. Page 3. Lines 86-90 and Figure 1: For Se level, please use one unit (either ng/mL or ug/L) for the sake of consistency.

Author Response

RESPONSE TO REVIEWER #3:

Thank you for your review of our manuscript entitled “Selenium as a bioactive micronutrient in the human diet and its cancer chemopreventive activity” for publication in Nutrients. Overall, we find Reviewer suggestions to be helpful, and constructive, and the corresponding revisions have strengthened the paper in multiple ways. According to your suggestions, we have expanded our manuscript with the content you suggested (the changes made in the manuscript are marked in red).

Major points:

  • “Please include a brief detail of dietary Se absorption and its transport.”

Response: Information has been added.

  • “Chapter 3.1.1. Glutathione peroxidases. Page 7: The role of GPX4 in ferroptosis (an iron-dependent cell death mechanism) has been completely ignored. Ferroptosis has emerged as an entirely new field during the last decade, as evident from innumerous publications. Please include a discussion of a reasonable length and content.”

Response: The manuscript mentions ferroptosis. However, as this phenomenon has been more extensively described in our previous publication (Radomska et al., Int J Mol Sci. 2021 Jan 20;22(3):1009) in order to avoid the accusation of self-plagiarism, we have not discussed this topic in a broad context in this paper.

Minor points:

  • “Section 2.1. Page 2. Line 69: “Richie et al. proved”. It would read better if ‘proved’ is replaced by either ‘observed’ or ‘showed’.”

Response: It has been changed.

  • “Section 2.1. Page 3. Lines 84-85 & Section 3.1.4. Page 9. Lines 249-251: Two examples are highlighted. Throughout the manuscript, number of sentences are written in this complex manner using multiple which or what. Please simplify the statements for convenient reading”

Response: It has been corrected.

  • Section 2.1. Page 3. Lines 86-90 and Figure 1: For Se level, please use one unit (either ng/mL or ug/L) for the sake of consistency.

Response: Units have been unified.

We hope that our answer will satisfy Reviewer and Editor. We thank you for consideration and await your decision.
